# Pancreas Rejection in the Artificial Intelligence Era: New Tool for Signal Patients at Risk

**DOI:** 10.3390/jpm13071071

**Published:** 2023-06-29

**Authors:** Emanuel Vigia, Luís Ramalhete, Rita Ribeiro, Inês Barros, Beatriz Chumbinho, Edite Filipe, Ana Pena, Luís Bicho, Ana Nobre, Sofia Carrelha, Mafalda Sobral, Jorge Lamelas, João Santos Coelho, Aníbal Ferreira, Hugo Pinto Marques

**Affiliations:** 1Hepatobiliopancreatic and Transplantation Center, Curry Cabral Hospital, Centro Hospitalar Universitário de Lisboa Central, R. da Beneficência 8, 1050-099 Lisbon, Portugal; inesfigueiredodebarros@gmail.com (I.B.); beatriz.chumbinho@gmail.com (B.C.); editefilipe@sapo.pt (E.F.); anapena64@gmail.com (A.P.); bicho.luis@gmail.com (L.B.); nobre.ana73@gmail.com (A.N.); sofia_carrelha@hotmail.com (S.C.); mafaldasnsobral@gmail.com (M.S.); jorgelamelas2000@yahoo.com (J.L.); jsantoscoelho71@hotmail.com (J.S.C.); hugoscpm@gmail.com (H.P.M.); 2Nova Medical School, Faculdade de Ciências Médicas, Universidade NOVA de Lisboa, 1169-056 Lisbon, Portugal; rita.s.ribeiro13@gmail.com; 3Blood and Transplantation Center of Lisbon, Instituto Português do Sangue e da Transplantação, Alameda das Linhas de Torres, n 117, 1769-001 Lisbon, Portugal; 4iNOVA4Health, Advancing Precision Medicine, RG11, Reno-Vascular Diseases Group, NOVA Medical School, Faculdade de Ciências Médicas, Universidade NOVA de Lisboa, 1169-056 Lisbon, Portugal; anibalferreira@icloud.com; 5Nephrology, Hospital Curry Cabral, Centro Hospitalar Universitário de Lisboa Central, R. da Beneficência 8, 1050-099 Lisbon, Portugal

**Keywords:** artificial intelligence, machine learning, pancreas transplantation, allograft survival, allograft rejection, patient risk management

## Abstract

Introduction: Pancreas transplantation is currently the only treatment that can re-establish normal endocrine pancreatic function. Despite all efforts, pancreas allograft survival and rejection remain major clinical problems. The purpose of this study was to identify features that could signal patients at risk of pancreas allograft rejection. Methods: We collected 74 features from 79 patients who underwent simultaneous pancreas–kidney transplantation (SPK) and used two widely-applicable classification methods, the Naive Bayesian Classifier and Support Vector Machine, to build predictive models. We used the area under the receiver operating characteristic curve and classification accuracy to evaluate the predictive performance via leave-one-out cross-validation. Results: Rejection events were identified in 13 SPK patients (17.8%). In feature selection approach, it was possible to identify 10 features, namely: previous treatment for diabetes mellitus with long-term Insulin (U/I/day), type of dialysis (peritoneal dialysis, hemodialysis, or pre-emptive), de novo DSA, vPRA_Pre-Transplant (%), donor blood glucose, pancreas donor risk index (pDRI), recipient height, dialysis time (days), warm ischemia (minutes), recipient of intensive care (days). The results showed that the Naive Bayes and Support Vector Machine classifiers prediction performed very well, with an AUROC and classification accuracy of 0.97 and 0.87, respectively, in the first model and 0.96 and 0.94 in the second model. Conclusion: Our results indicated that it is feasible to develop successful classifiers for the prediction of graft rejection. The Naive Bayesian generated nomogram can be used for rejection probability prediction, thus supporting clinical decision making.

## 1. Introduction

Personalized medicine and precision medicine, which are complementary approaches that work together to improve patient outcomes by providing more targeted and effective medical care, form a holistic approach to health care that enables for personalized diagnosis and treatment [1].

While genetic testing and other OMICS-based personalized medicine approaches can be expensive, machine learning, a branch of artificial intelligence, can be a useful tool for personalized medicine. Machine learning can analyze conventional clinical data that are already being collected as part of routine patient care. Its growing importance in personalized medicine enables healthcare providers to analyze and interpret large amounts of data, leading to the identification of patterns and associations that may not be visible to human perception. This can potentially lead to individualized or tailored treatments and diagnoses [2].

Simultaneous pancreas–kidney (SPK) transplantation is a viable treatment option for patients with end-stage renal disease and type 1 diabetes [3]. In recent years, personalized medicine has become increasingly important in the context of solid organ transplantation [4]. For instance, Sirota et al. [5] reviewed this topic and discussed the potential of using omics technologies to improve solid organ transplant outcomes and move toward personalized treatments in transplantation research. The authors described how various omics technologies, including genomic, transcriptomic, proteomic, and metabolomic studies, could lead to identifying biomarkers and molecular pathways important for transplant success or failure. In this review, the challenges and potential applications associated with these technologies were also highlighted (e.g., standardization of methods), thus resulting in more readable results between transplantation centers. Overall, they argue that the use of omics technologies has the potential to revolutionize transplantation research and improve transplant outcomes, making the field of transplantomics an important area of research in the coming years. More recently, Maldonado et al. [4] highlighted the importance and impact of the most recent developments in personalized medicine and noninvasive diagnostic techniques and how they have impacted the outcomes in solid organ transplantation, which were mainly summarized by the limitations of current immunosuppressive therapies and the importance of individualizing treatment regimens. The paper emphasizes the potential for personalized medicine and noninvasive diagnostics in solid organ transplantation and the need for continued research and collaboration to fully realize these benefits for transplant patients.

Additionally, with the boom in the application of artificial intelligence (AI) to everything and everyone, healthcare and transplantation in particular having been impacted by AI, with its application in clinical decision-making, biomedical research, and medical education. Indeed, several researchers have focused their work on the applicability of these new tools as a means to better understand transplant biology and improve outcome results from transplantation. Works as presented by Mekov [6] in lung transplantation, Xu et al. [7] and Piening et al. [8] in heart transplantation, Yu et al. [9] in liver transplantation, or Thongprayoon et al. [10], Kazi et al. [11], and Yoo et al. [12] in kidney transplantation, all the authors as well as others have tried to use AI tools as a means to either evaluate waiting lists and access to transplants or evaluate outcomes. The introduction of these methods by lead several regulatory entities, e.g., Food and Drug Administrations indicate that they felt the need to intervene. Validating and approving tools have also been introduced, and these are either devices or products that can imitate intelligent behavior or mimic human learning and reasoning [13].

Additionally, in SPK transplantation, researchers have been exploiting ways in which personalized medicine can benefit patients and transplant outcomes. As an example, they have explored ways to use genetic and other biomarkers to identify patients who are most likely to benefit from the procedure and to optimize immunosuppressive therapy after transplantation to minimize the risk of rejection and other complications. For example, measuring levels of certain immune cell subsets, such as regulatory T cells or B cells, may help identify patients who are at higher risk of acute rejection or infection after transplantation [14]. Another example of a personalized medicine approach in SPK transplantation is the use of pharmacogenomics, which is the study of how genetic variation affects drug response. Some patients may have genetic mutations that influence how certain drugs are metabolized, resulting in harmful side effects or suboptimal efficacy. By tailoring medications appropriately, clinicians may be able to optimize treatment results, thus minimizing the danger of unfavorable outcomes [15,16].

However, allograft rejection remains a major concern in solid organ transplantation, and there is a need for less invasive diagnostic tools and more targeted and effective treatments. Personalized medicine and machine learning offer potential solutions to these challenges. In SPK transplantation, as in other solid organ transplants, machine learning can be used to predict allograft rejection and develop personalized treatment plans. This approach involves the analysis of a patient’s genetic, clinical, and environmental factors to identify the optimal treatment plan. By using personalized medicine, clinicians can better predict which patients are at risk of rejection and provide more targeted treatments. For example, machine learning algorithms can be used to predict which patients are at risk of allograft rejection based on their clinical and genetic characteristics. These predictions can help clinicians develop personalized treatment plans that are tailored to each patient’s needs.

In this study, our aim was to evaluate the feasibility of identifying features and risk factors, based on conventional laboratory analysis, both pre- and post-operatively and during the follow-up period. Additionally, we aimed to develop a machine learning-based model that could measure the likelihood of dysfunction or pancreas allograft rejection. This model would enable clinicians to take appropriate actions to optimize and improve pancreatic graft survival.

## 2. Materials and Methods

### 2.1. Study Population

This retrospective cohort study was conducted at Hospital Curry Cabral-Centro Hospitalar e Universitário de Lisboa Central. We identified all recipients who underwent technically successful simultaneous pancreas–kidney (SPK) transplants from systemic-drained, whole-organ brain-dead donors between March 2011 and January 2020, resulting in a total of 106 patients. However, 33 patients were excluded from the study (refer to Figure 1). Therefore, a final cohort of 73 participants was included and evaluated until March 2021. The study received approval from the CHULC ethics committee (number 985/2020). Patient data were collected from the SClinic database (Hospital Clinical System), an evolving information system developed by the Ministry of Health for clinical digitization within the National Health Service. It aims to standardize clinical record procedures to ensure information normalization. All patients provided informed consent to participate in the study. The procurement operation, back-table bench procedures, and recipient surgery were previously described in detail by our research group in a separate publication [17].

### 2.2. Immunosuppression Protocol

To prevent rejection of transplanted organs, T cell-depleting antibodies were administered 2–4 h before transplant surgery, e.g., polyclonal rabbit antithymocyte globulin (rATG-Thymoglobulin), at a dosage of 1.5 mg/kg and, whenever needed, continued intraoperatively. This was followed by 1.5–2 mg/kg of rATG-Thymoglobulin per day during the six days after surgery, for a total of seven doses. However, if the patient’s white blood cell count was less than 2000/microL and/or the platelet count was less than 75,000/microL, the rATG-Thymoglobulin dose was adjusted accordingly. For the first three days after transplantation, three 500 mg IV methylprednisolone injections were given, with the first one before surgery and the remaining two on the first and second days after surgery. After the third day, oral prednisone was administered at a dose of 20 mg once daily for 2–3 weeks, which was then tapered down to 5 mg once daily by the third month. Mycophenolate mofetil (MMF) was given orally at a dose of 500 mg before surgery. After surgery, it was administered at a dose of 250 mg intravenously twice daily until day 5, after which it was switched to 360 mg of enteric-coated mycophenolate sodium (ECMPS) which was ingested orally twice daily. In case of acute cellular rejection, we treated patients with three daily pulses of 500 mg methylprednisolone, followed by 1.5 mg/kg of rATG-Thymoglobulin. For antibody-mediated rejection, our primary goal was to eliminate the clonal population of B cells or plasma cells that produce the donor-specific antibody (DSA). To achieve this, we performed five plasmapheresis sessions followed by intravenous immune globulin (IVIG) and a 500 mg Rituximab IV infusion after the fifth plasmapheresis session.

### 2.3. Rejection Classification and Identification

The preferred method for diagnosing and grading pancreas transplant rejection is through biopsy, but it is not commonly performed due to the risk of complications. In simultaneous pancreas–kidney (SPK) transplants, a kidney biopsy is usually conducted first, which may be sufficient to justify rejection treatment [18]. Considering the avoidance of pancreas biopsies and the primary reliance on laboratory testing for rejection monitoring, clinical indicators that suggest rejection were used to investigate the relationship with dd-cfDNA values. These indicators include a serum creatinine increase of more than 30% and/or lipase levels exceeding site-specific normal limits in the presence of unexplained fever or leukocytosis, new graft tenderness, or abnormal endocrine results such as fasting glucose levels. If clinically necessary, for-cause biopsies were performed according to standard care protocols. Pancreas and kidney functions were evaluated at each post-operative visit and through regular laboratory screenings. We recorded the number of pancreas and renal transplant biopsies conducted, as well as the clinical reasons for performing them [19].

### 2.4. Features and Data Analysis

A total of 74 clinical features were extracted from electronic patient and donor records. Among these, 30 were categorical features related to the recipient, including gender, renal replacement therapy (hemodialysis, peritoneal dialysis, or pre-emptive), de novo DSA, American Society of Anesthesiology Classification (ASA), and others. The remaining 44 features were numeric and included age at the date of transplant, weight, height, duration of renal replacement therapy, duration of diabetes mellitus, previous treatment for diabetes mellitus with long-term insulin (U/I per day), SUM pre-transplant DSA MFI (maximum), vPRA Pre-Tx (%), SUM pre-transplant DSA MFI (latest assay), and more.

Donor features encompassed age, weight, height, intensive care unit days, pancreas donor risk index (pDRI) for numeric features, and cause of death and cardiac arrest for categorical features. Surgical features were also included, such as intraoperative blood transfusion, total operating time, cold and warm ischemia time, and others.

Machine learning algorithms (ML) and feature selection were performed in Orange 3 version 3.19.0 (Bioinformatics Lab, University of Ljubljana, Slovenia). Supervised ML algorithms were assessed by area under the receiver operating characteristics (AUCs) and classification accuracy (CA); F-1 score and precision and sensitivity were also calculated. Data features visualization was performed via unsupervised learning dimensionality reduction using t-Distributed Stochastic Neighbor Embedding (t-SNE).

AUC and CA are two methods available to evaluate the effectiveness of a classification model. The AUC curve considers two factors, namely, the True Positive Rate and False Positive Rate. On the other hand, classification accuracy measures the percentage of correctly classified subjects. To assess the model’s performance, a leave-one-out cross-validation approach was used. This method involves using all available data points except for one, which is used to train the model, and then using the omitted data point to test the model. This process is repeated for each data point in the dataset, and the performance of the model is averaged across all the iterations.

Feature selection was conducted using an information gain algorithm. This selection method is used to identify the most relevant features for a particular classification problem. It is calculated by computing the entropy of the target variable before and after the split, and the difference is the information gain. In the end, features with high information gain are considered to be more relevant to the classification problem. The selection resulted in the identification of ten features (two categorical and eight numeric) which best contributed to the strength of the models, namely: previous treatment for diabetes mellitus with long-term insulin (U/I/day), type of dialysis (peritoneal dialysis, hemodialysis, or pre-emptive), de Novo DSA, vPRA_Pre-Transplant (%), donor blood glucose, pancreas donor risk index (pDRI), recipient height, dialysis time (days), warm ischemia (minutes), recipient of intensive care (days).

### 2.5. Algorithms

Several ML algorithms were tested in the end based on their performance and AUC, and CA Naive Bayes and Support Vector Machine (SVM) classifier algorithms were used.

#### 2.5.1. Naive Bayes

Naive Bayes is a probability classification algorithm that uses Bayes’ theorem to make predictions based on the probability of a particular event occurring given input data. It assumes that all features of the input data are independent of each other, which is called the ‘‘naïve” assumption. Notwithstanding this assumption, Naive Bayes has been highly used and has provided evidence of good performance in lots of real-world applications, including healthcare. It works by calculating the probability of a data point being associated with a specific class by multiplying the probabilities of each feature given that class, choosing the class with the highest probability as the prediction. Given its simplicity it can be useful in situations where the independence assumption is reasonably accurate or where the interdependence between features is not essential [20].

#### 2.5.2. Support Vector Machine

The Support Vector Machine is a supervised machine learning algorithm that is commonly used for classification and regression analysis. The SVM works by finding the optimal hyperplane in a high-dimensional space that maximally separates the data points of different classes. The hyperplane is defined by a set of parameters called weights, and the goal of the algorithm is to find the optimal set of weights that maximizes the margin between the hyperplane and the nearest data points of each class. The SVM has been applied in various clinical scenarios, such as medical image analysis, disease diagnosis, and outcome prediction, where it can learn to classify patients based on features extracted from their data [20].

#### 2.5.3. t-Distributed Stochastic Neighbor Embedding

t-Distributed Stochastic Neighbor Embedding is a dimensionality reduction technique, mostly useful in visualizing complex, high-dimensional datasets, and is particularly effective in preserving the underlying structure and relationships in high-dimensional data as it can preserve both local and global structures. The algorithms work by modeling the similarity among pairs of statistical points within the high-dimensional area and the low-dimensional area, aiming to hold the neighborhood relationships of the statistics points as much as possible. This set of rules may be precious in medical studies for visualizing affected person clusters or figuring out subgroups inside a population [21]. The t-SNE algorithm first constructs a probability distribution that defines the similarity between the high-dimensional data points based on their Euclidean distances.

## 3. Results

The t-SNE score plot presented in Figure 2, based on all 74 features, shows that no major cluster formation can be observed. This result was somewhat expected since a larger number of features in relation to the number of patients can lead to the identification of a feature pattern unique to each patient instead of being unique to the analyzed condition.

Likewise, when more complex algorithms are employed, such as the SVM and Naive Bayes algorithm presented in Table 1, the results are disappointing. In the SVM model, the AUC and CA values are 0.70 and 0.77, respectively, indicating satisfactory–good performance. However, the specificity for rejection detection is quite low at 46%. The Naive Bayes model shows some improvement in terms of specificity (90%) and AUC (0.89), but the overall CA is very poor, at 0.47.

To address this problem and identify the features that had the greatest impact on the predictive model, a feature selection algorithm was employed. In this particular case, the information gain algorithm was used.

As observed in Figure 3, information gain improved the t-SNE plots, forming clusters with the t-SNE model (Figure 3A), SVM model (Figure 3B), and Naive Bayes model (Figure 3C). In the case of the t-SNE plot, there were seven misclassified patients (two patients with rejection in the non-rejection group, and five non-rejection patients in the rejection group). However, these misclassifications did not result in any significant worsening of the outcomes. Nonetheless, the models themselves showed significant improvement, as shown in Table 2, with an AUC and CA of 0.96 and 0.94, respectively, for the SVM model, and 0.97 and 0.87 for the Naive Bayes model. There was also a major improvement in terms of specificity, particularly for the SVM model, which increased from 0.46 (using all features) to 0.93 (using the top 10 features identified using information gain). The AUC performances of the models (Figure 4) were consistent for both the classification of rejection and non-rejection cases. As a result, the SVM model correctly predicted 93.9% of true rejection patients and 92.3% of true non-rejection patients, with a 13.8% false classification rate. The Naive Bayes model had a 15.2% false classification rate (all predicted as rejection patients when they were not), but it achieved a 100% prediction rate for all patients classed as rejection patients (Figure 5).

Regarding the features that contribute the most to the models’ AUC and CA (Figure 6), the type of kidney replacement treatment (peritoneal dialysis, hemodialysis, or pre-emptive) was found to be one of the most influential features for the models’ AUC results. It ranked first for the SVM model and second for the Naive Bayes model. The formation of de novo DSA also emerged as an important feature, appearing in the top three features for both models. However, in terms of CA, the type of kidney replacement treatment had a negative impact on the Naive Bayes model (Figure 6D), while it was the second most important feature for the SVM model. It is worth noting that 5 out of 10 features had a negative impact on the CA of the Naive Bayes model, which could partially explain the lower value obtained.

As the Naive Bayes model provides a good AUC and CA, we then constructed a nomogram plot to predict pancreas rejection in SPK transplantation, as shown in Figure 7, based on clinical and laboratory features. This nomogram-based strategy has been used in other types of transplants, e.g., kidney [22], lung [23], and heart [24], as a means to predict transplant-related events. In our model the 10 top features identified using information gain were incorporated, and our endpoint was the ability to identify or generate risk stratification for pancreas rejection in SPK transplantation, and to our knowledge this is one of the first attempts of model-based risk stratification for pancreas rejection.

## 4. Discussion

Recent advancements in ML have shown promising results in the field of medical decision making. In particular, ML algorithms have the potential to support clinical decisions by identifying patients at risk of rejection after transplantation [25].

However, before the implementation of these new methodologies to support clinical medical decision making, several questions must be answered. Can we, based on clinical data, only identify patients that are more prone to have a rejection event? Are all features contributing to the model? If not, which are the most informative features that better contribute to the model?

To answer the first question, it is possible to use clinical data to identify patients who may be more prone to rejection events. In healthcare, ML algorithms can be trained on large datasets of patient information and outcomes to identify patterns and factors that are associated with several healthcare events and scenarios [26]. In the field of transplantation and in pancreatic transplants, several authors have developed algorithms to make predictions of which patients are at a higher risk for rejection of pancreatic islet grafts [27] or of suffering delayed pancreatic endocrine graft functioning.

Regarding the second question, not all features will necessarily contribute equally to the model. Some characteristics may be less important or even irrelevant or can even result in worse results, since the expanded complexity as a result of the added number of features several times over has no major impact on the prediction values. Many times, it is better to simplify the model to improve its accuracy [28]. Thus, and answering the last question to determine which features have the most information and contribute the most to the model, a feature importance analysis can and should be performed. In our study on information gain, as expected we observed a major gain in terms of AUC and CA [29].

Our improved models, based on pre-transplant recipient, donor, and transplantation features, have shown that they can be practical tools for identifying patients at risk of rejection after transplantation. It is highly desirable to identify this risk early in the immediate post-transplant period so that we can take measures to minimize inflammatory/immune aggression on the pancreatic graft. Therefore, we identified the features that had the greatest impact on the models, some of them already identified in the literature for other solid organs, namely those related to: (1) pre-transplant recipient factors: pre-transplant insulin requirement and virtual panel reactive antibody (VPRA) percentage; traditionally, a high vPRA has been associated with increased risk of rejection [30]; and height, type, and length of dialysis, which are significant factors in determining the onset of graft function and rejection [31]. (2) Donor factors: pancreas donor risk index (PRDI); in the literature for the PRDI, 1.24 or higher donor grafts had significant poorer outcome compared to a PDRI less than 1.24 [32]; blood glucose level. (3) Transplantation factor: warm ischemia time. (4) Immediate post-transplant factors: days in intensive care and de novo donor-specific antibodies (DSA). De novo DSA against both classes I and II HLA conferred poorer graft survival [33].

The provided results discuss the application of machine learning algorithms, specifically Support Vector Machines (SVMs) and Naive Bayes, for the identification of rejection in simultaneous kidney–pancreas transplantation. Initially, when all 74 features were considered, the t-SNE score plot did not show any significant cluster formation, indicating that individual feature patterns were unique to each patient rather than being unique to the analyzed condition. Similarly, the performance of the SVM and Naive Bayes models based on all features was not satisfactory, with lower specificity and overall accuracy. To address this issue and identify the features with the greatest impact on the predictive model, a feature selection algorithm called Info Gain was employed. The application of Info Gain resulted in improved t-SNE plots, forming clusters with the t-SNE model, SVM model, and Naive Bayes model. Although there were some misclassifications, they did not significantly worsen the outcomes. The AUC and overall accuracy values improved significantly for both the SVM and Naive Bayes models after feature selection.

Specifically, the SVM model achieved an AUC of 0.96 and an overall accuracy of 0.94, with a substantial increase in specificity from 0.46 to 0.93 when using the top 10 features identified using Info Gain. The Naive Bayes model also showed improvement, with an AUC of 0.97 and an overall accuracy of 0.87. However, the Naive Bayes model had a higher false classification rate compared to the SVM model.

The analysis of feature importance revealed that the type of kidney replacement treatment and the formation of de novo DSA were among the most influential features for the models’ AUC results. However, the impact of these features on the overall accuracy (CA) differed between the SVM and Naive Bayes models. As a result, the SVM model correctly predicted rejection in 93.9% of true rejection patients and non-rejection in 92.3% of true non-rejection patients, with a 13.8% false classification rate. The Naive Bayes model had a 15.2% false classification rate, but it achieved a 100% prediction rate for all patients with rejection.

Based on the promising performance of the Naive Bayes model, a nomogram plot was constructed to predict pancreas rejection in simultaneous kidney–pancreas transplantation. The nomogram incorporated the top 10 features identified using Info Gain and aimed to provide a means of predicting or risk stratifying pancreas rejection in this transplantation setting and has the potential to serve as a useful tool in supporting medical decision making, particularly for patients for whom undergoing a biopsy can be risky.

In the future, this nomogram could provide valuable assistance in clinical settings, particularly in the clinical decision-making process when taking measures that could preserve the transplanted pancreas, improve patients’ quality of life, and avoid missing the opportunity for transplantation due to organ scarcity and the lower likelihood of retransplantation in case of graft loss. Based on the findings of the study, there are several potential future research directions that can build upon the current work and expand the application of machine learning models in transplantation settings. These directions are as follows: (a) Conducting external validation studies using independent datasets from different transplantation centers and diverse patient populations would be essential to validate the developed machine learning models. External validation helps assess the models’ generalizability and ensures their reliability and effectiveness across different settings. (b) Conducting prospective studies that collect data in real-time would provide more robust evidence for the models’ predictive capabilities. Prospective studies allow for more accurate and standardized data collection, minimizing biases associated with retrospective designs. (c) Assessing the applicability and generalizability of machine learning models in other transplantation settings beyond simultaneous kidney–pancreas transplantation is crucial. The models can be tested and validated in different solid organ transplantations, such as liver, heart, or lung transplants, to determine their utility and potential for personalized rejection prediction in those contexts. (d) Integration with Clinical Decision Support Systems, exploring the integration of machine learning models into clinical decision support systems to facilitate their practical implementation in transplantation clinics. By integrating the models into electronic health record systems, clinicians can receive real-time predictions and recommendations for personalized patient management, aiding in the decision-making process and potentially improving patient outcomes.

By pursuing these future research directions, the field can advance the understanding and application of machine learning models in transplantation, ultimately leading to more personalized and effective patient care, improved outcomes, and enhanced decision-making processes for transplant clinicians.

This study has some limitations and biases; the study focused on a specific set of clinical and laboratory features, resulting in a limited number of variables being considered for analysis. There may be other relevant factors not included in the study that could influence the prediction of rejection in SPK transplantation. Additionally, there may be missing data for some variables, which could introduce bias and affect the robustness of the models. The preferred method for diagnosing and grading pancreas transplant rejection is through biopsy. However, due to the potential risks and complications associated with biopsies, they are not commonly performed. As a result, the reliance on clinical indicators and laboratory testing for rejection monitoring introduces limitations in the accuracy of rejection classification and identification.

## 5. Conclusions

Machine learning and personalized medicine can potentially provide answers to these problems. Machine learning algorithms can predict the likelihood of rejection by looking at a patients clinical and transplant-related factors, allowing clinicians to take proactive steps to stop rejection before it happens or to discriminate between whom needs to undergo a biopsy or not, thus creating an individualized follow-up plan that caters to the unique requirements of each patient.

We support the development of less-invasive diagnostic tools, such as the presented nomogram, and more specialized therapies for pancreas transplant allograft rejection. In this regard, personalized medicine and machine learning hold great promise, providing clinicians with the ability to anticipate rejection and create tailored treatment regimens based on individual patient and donor factors. This approach could significantly enhance the long-term success of pancreas transplantation. In recent years, there has been substantial advancement in the field of organ transplantation, with pancreatic transplantation becoming a critical therapy option for people with type 1 diabetes. However, one of the biggest difficulties faced by clinicians is the potential for allograft rejection, which can result in transplant failure and the need for additional interventions.

In conclusion, in this research, we stress the significance of creating less invasive diagnostic tools and more specialized therapies for pancreas transplant allograft rejection. Currently, to diagnose rejection, clinicians rely on invasive biopsy procedures, which can be unpleasant and risky. As a result, new diagnostic approaches that can effectively detect or identify patients at risk of rejection are required. These techniques should be able to provide patient stratification allowing for the reduction of biopsy risk-associated events.

## Figures and Tables

**Figure 1 jpm-13-01071-f001:**
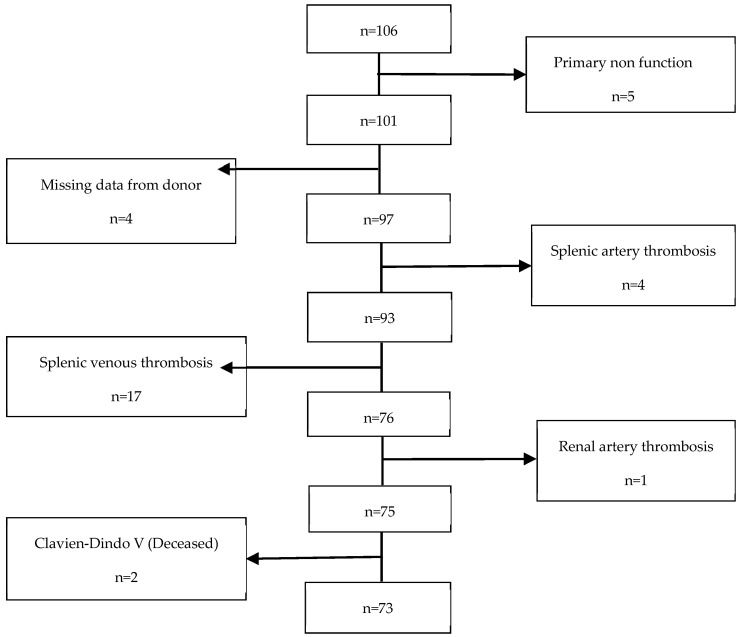
Exclusion criteria.

**Figure 2 jpm-13-01071-f002:**
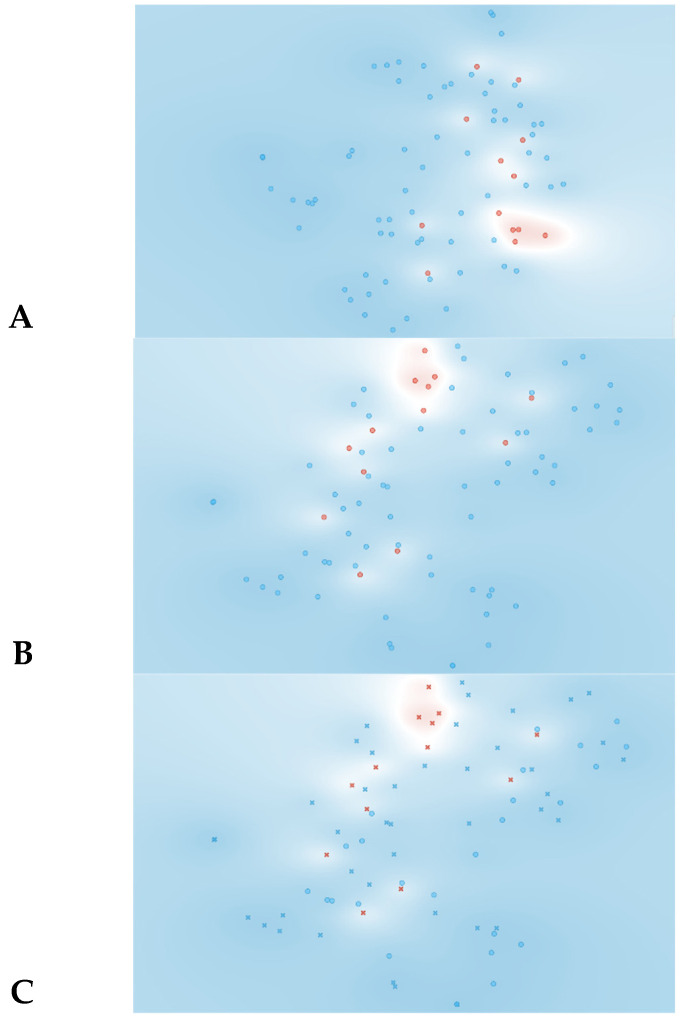
Data visualizations in 2D using t-SNE for all features (*n* = 74). (**A**) t-SNE all features, (**B**) t-SNE plot of SVM algorithm, (**C**) t-SNE plot of Naive Bayes algorithm. Red represents yes for rejection; blue represents no for rejection. In plots B and C circles represent algorithm classification as rejection and cross represents classification as non-rejection.

**Figure 3 jpm-13-01071-f003:**
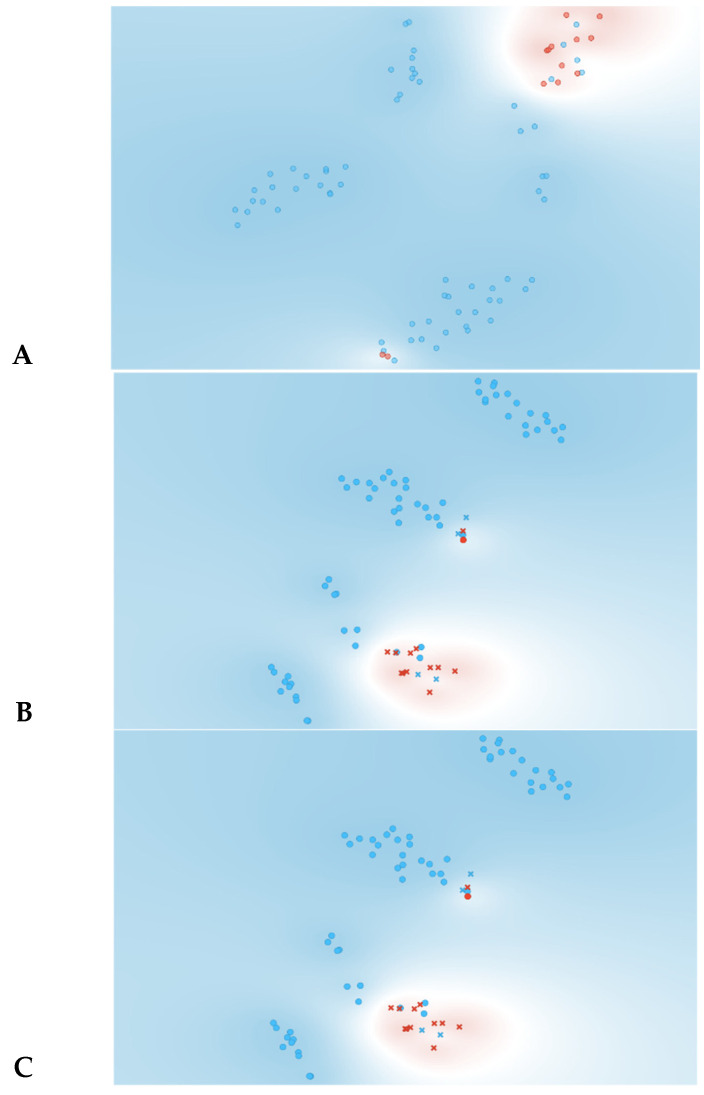
Data visualizations in 2D using t-SNE form features selection (*n* = 10). (**A**) t-SNE all features, (**B**) t-SNE plot of SVM algorithm, (**C**) t-SNE plot of Naive Bayes algorithm. Red represents yes for rejection; blue represents no for rejection. In plots (**B**,**C**) circles represent algorithm classification as rejection and cross represents classification as non-rejection.

**Figure 4 jpm-13-01071-f004:**
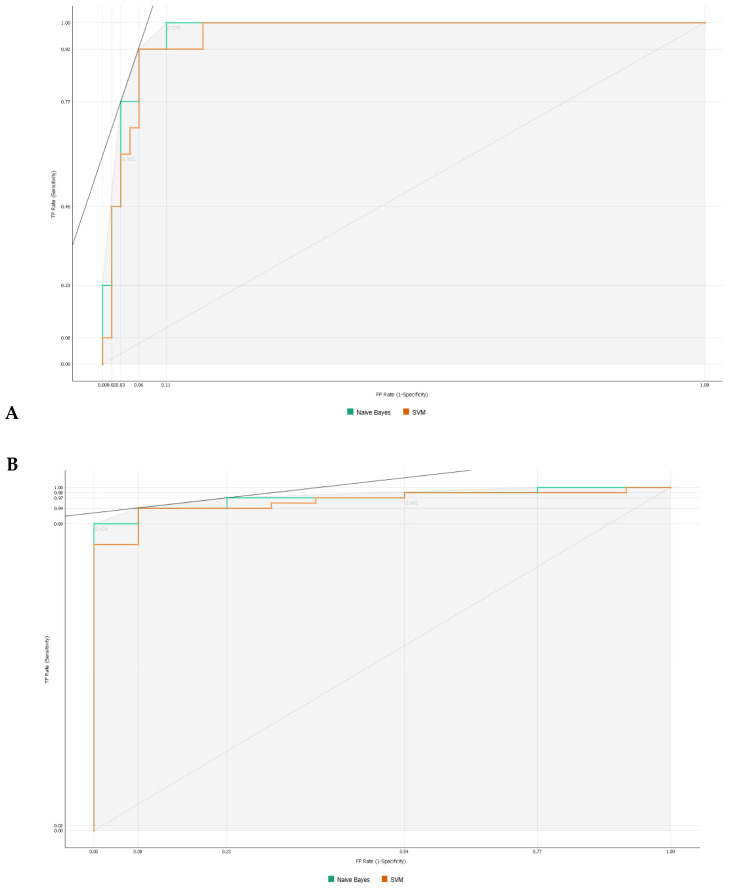
Area under the receiver operating characteristics for machine learning algorithms. (**A**) Classification as rejection. (**B**) Classification as non-rejection.

**Figure 5 jpm-13-01071-f005:**
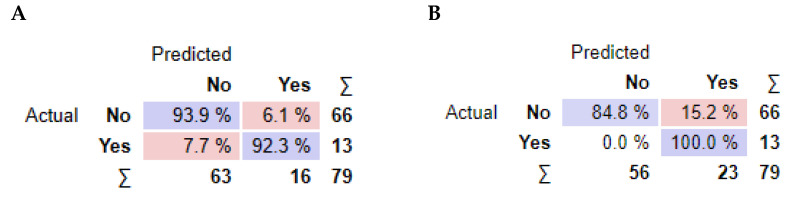
Performance summary of the classification models. For each model. average values of confusion matrix cells are reported. (**A**) Support Vector Machine algorithm, (**B**) Naive Bayes algorithm.

**Figure 6 jpm-13-01071-f006:**
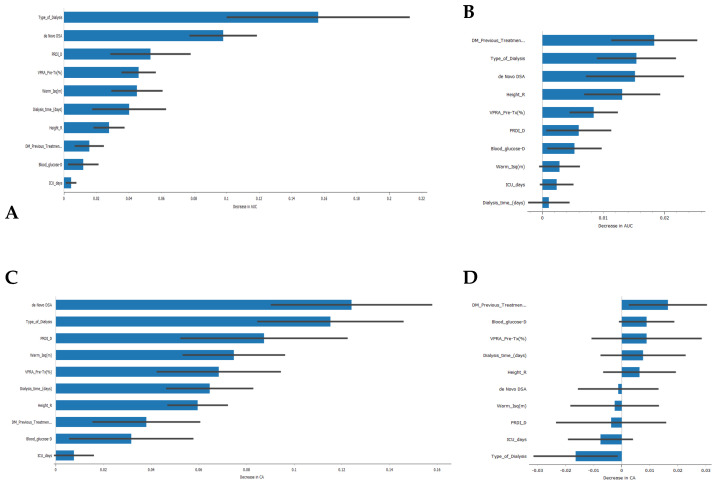
Feature importance in the development of classification models. (**A**) Feature importance in the Support Vector Machine algorithm for the AUC result, (**B**) feature importance in the Naive Bayes algorithm for the AUC result, (**C**) feature importance in the Support Vector Machine algorithm for the CA result, (**D**) feature importance in the Naïve Bayes algorithm for the CA result.

**Figure 7 jpm-13-01071-f007:**
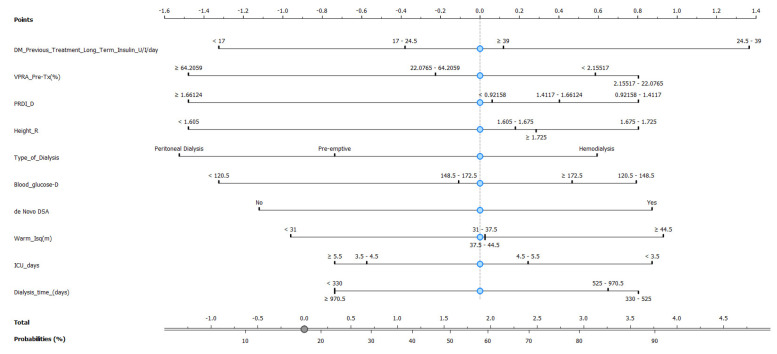
Nomogram plot based on Naive Bayes model to predict pancreas rejection in simultaneous kidney–pancreas transplant.

**Table 1 jpm-13-01071-t001:** Machine learning algorithm models for identification of rejection in simultaneous kidney pancreas transplantation based on all features (*n* = 74).

Model	AUC	CA	F1	Precision	Recall	Specificity
SVM	0.70	0.77	0.78	0.79	0.77	0.46
Naive Bayes	0.89	0.47	0.51	0.87	0.47	0.90

SVM, support vector machines; AUC, area under the receiver operating characteristics; CA, classification accuracy; F1, F1-score.

**Table 2 jpm-13-01071-t002:** Machine learning algorithm models for identification of rejection in simultaneous kidney pancreas transplantation based on all features (*n* = 10).

Model	AUC	CA	F1	Precision	Recall	Specificity
SVM	0.96	0.94	0.94	0.95	0.94	0.93
Naive Bayes	0.97	0.87	0.89	0.93	0.87	0.98

SVM, support vector machines; AUC, area under the receiver operating characteristics; CA, classification accuracy; F1, F1-score.

## Data Availability

The datasets used and/or analyzed during the current study are available from the corresponding author upon reasonable request.

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
