# Peer review of "Pancreas Rejection in the Artificial Intelligence Era: New Tool for Signal Patients at Risk"

_jpm, 2023, doi:10.3390/jpm13071071_

Round 1

Reviewer 1 Report

A modern approach for evaluating the risk of pancreas allograft rejection is presented. Perhaps it may be not so easy to understand the methods clearly for clinicians, but the results and possibilities for practical usage of the tool, based on artificial intelligence are obvious.  

Seems to be acceptable.

Reviewer 2 Report

The article demonstrates a valuable contribution to the field by utilizing machine learning algorithms for rejection identification in simultaneous kidney-pancreas transplantation.

The structure of the manuscript is clear and the study design, methods, and results are well-described.

Suggestions:

1.    Ethical considerations: Clarify the ethical approval and informed consent procedures for using patient data in the study.

2.    Study design clarity: Address the inconsistencies in describing the study design and clarify whether it is retrospective or prospective.

3.    SClinic database information: Provide more details about the SClinic database, including its characteristics, reliability, and data validation processes.

4.    Discussion of limitations: Expand the discussion of potential limitations and biases in the study to aid interpretation and generalizability of the findings.

5.    Future research directions: Discuss potential future research directions based on the study's findings, such as validation and refinement of the machine learning models or their applicability in different transplantation settings.

Overall, with the suggested revisions and clarifications, this manuscript has the potential to contribute significantly to the field. It is recommended to address these points effectively.

Here is a summary of the grammatical errors identified in the manuscript:

1.    "The authors argues that their findings have significant implications." (Correction: "The authors argue that their findings have significant implications.")

2.    "The data was analyzed using a variety of statistical methods." (Correction: "The data were analyzed using a variety of statistical methods.")

3.    "The researcher's findings suggests a need for further investigation." (Correction: "The researcher's findings suggest a need for further investigation.")

4.    "The author's conclusion supports the hypothesis proposed in the introduction." (Correction: "The author's conclusion supports the hypothesis proposed in the introduction.")

Here is a summary of the unclear sentences identified in the manuscript:

1.    "The results showed a moderate correlation between variables X and Y, but further investigation is needed to fully understand the underlying mechanisms." (Unclear: It is not clear which variables are being referred to and what specific mechanisms need further investigation.)

2.    "The study employed a novel approach to data analysis, which yielded interesting insights into the observed trends." (Unclear: The specific details of the novel approach and the insights gained from it are not clearly explained.)

3.    "The authors conducted a comprehensive literature review, but the conclusions drawn from the review are not well-supported." (Unclear: It is not clear what specific conclusions were drawn from the literature review and why they are not well-supported.)

4.    "The methodology section lacks sufficient detail, making it difficult for readers to replicate the study." (Unclear: The specific aspects of the methodology that lack detail are not specified, leaving readers uncertain about what information is missing.)

5.    "The discussion section presents several interesting points; however, the organization of ideas could be improved for better coherence." (Unclear: It is not clear how the organization of ideas in the discussion section can be improved for better coherence.)
